# Formation and Characterization of Lactoferrin-Hyaluronic Acid Conjugates and Their Effects on the Storage Stability of Sesamol Emulsions

**DOI:** 10.3390/molecules23123291

**Published:** 2018-12-11

**Authors:** Runhua Liu, Jinhong Zhang, Caicai Zhao, Xiang Duan, David Julian McClements, Xuebo Liu, Fuguo Liu

**Affiliations:** 1Beijing Advanced Innovation Center for Food Nutrition and Human Health, Beijing Technology and Business University (BTBU), Beijing 100048, China; myasdzxc@163.com; 2College of Food Science and Engineering, Northwest A&F University, Yangling 712100, China; ZhangJH_C@163.com (J.Z.); zhaocaicai1994@163.com (C.Z.); duanxiang402@163.com (X.D.); 3Department of Food Science, University of Massachusetts Amherst, Amherst, MA 01003, USA; mcclements@foodsci.umass.edu

**Keywords:** lactoferrin, hyaluronic acid, covalent complex, sesamol emulsion, stability

## Abstract

The purpose of this study was to fabricate biopolymer conjugates from lactoferrin (LF) and hyaluronic acid (HA) and then to investigate their potential as emulsifiers for forming sesamol-loaded emulsions. Initially, LF-HA covalent conjugates were formed using the carbodiimide coupling method in aqueous solutions at pH = 4.5, and then the nature of the conjugates was investigated using sodium dodecyl sulfate-polyacrylamide gel electrophoresis (SDS-PAGE), Fourier Transform Infrared Spectroscopy (FTIR) spectroscopy, and fluorescence spectroscopy. The results demonstrated the formation of an amide link between the amine groups of LF and the carboxyl groups of HA. Sesamol emulsions were prepared using the LF-HA conjugates as emulsifiers and their stability was determined. The conjugates improved both the physical and chemical stability of the emulsions during storage. Optimum stability of the emulsion was obtained at a LF-to-HA molar ratio of 2:1. Our results suggest that LF-HA conjugates may be effective emulsifiers for use in food stuffs and other applications.

## 1. Introduction

Manipulating the interactions between proteins and polysaccharides can be used to modify the physicochemical and nutritional properties of many processed foods, as well as to design colloidal delivery systems for bioactive agents [1,2,3]. Many proteins are amphiphilic molecules capable of adsorbing to the surfaces of oil droplets and protecting them against aggregation, whereas many polysaccharides are extended hydrophilic molecules that can thicken or gel the aqueous phases of emulsions thereby improving their stability to gravitational separation [4]. Numerous studies have shown that the stability of emulsions can be improved by modulating their interfacial properties through the formation of protein-polysaccharide complexes [5,6]. The overall stability of these emulsions depends not only on the structure and properties of the individual ingredients, but also on the strength of the protein–polysaccharide interactions [4]. Protein–polysaccharide covalent conjugates have been shown to be highly effective for the oral delivery of nutraceuticals and drugs [2].

Sesamol is a phenolic component of sesame seed oil, which can inhibit lipid peroxidation, hydroxyl radical-induced deoxyribose degradation, and DNA cleavage [7]. Sesamol also exhibits strong antioxidant activity and is regarded as a very effective antioxidant in roasted sesame seed oil [8]. Its antioxidant activity is primarily attributed to its unique molecular structure: a phenolic group attached to a benzodioxole group [9]. However, sesamol also has some undesirable attributes that limit its potential application in foods and other commercial products. It has poor water-solubility, poor chemical stability (oxidation and photodegradation), limited oral bioavailability, and rapid elimination (as conjugates) from the body [10]. Even though sesamol is generated from sesamolin during the roasting or bleaching of sesame seed oil, it is not very stable to heat treatment. For instance, sesamol dissolved in methyl linoleate has been shown to degrade during heating at 180 °C for 60 min, with less than 40% remaining by the end of this process [11].

Sesamol has also been shown to degrade much faster when it is dispersed in aqueous solutions than in nonpolar media [12]. There is therefore interest in improving the dispersability and stability characteristics of sesamol by encapsulating in colloidal delivery systems, such as emulsions.

In the present study, lactoferrin (LF) and hyaluronic acid (HA) were selected as food-grade polymers to create biopolymer conjugates suitable for use as emulsifiers in sesamol-based nanoemlusions. LF is a glycoprotein with strong surface and antioxidant activities, as well as the ability to form both physical complexes and covalent conjugates with polysaccharides [13]. HA is a natural polysaccharide that can also be used to form complexes and conjugates [14]. In particular, the carboxyl group on the glucuronic acid (C-5) residues of HA, is an ideal candidate for covalent conjugation with amide and hydroxyl groups through amidification and esterification, respectively [15,16]. Previous studies have already shown that modified HA can be used in emulsion-based delivery systems [17].

In this study, LF-HA conjugates were synthesized through a two-step process. First, 1-(3-dimethylaminopropyl)-3-ethylcarbodiimide hydrochloride (EDC) and NHS were added to the HA solution and reacted at room temperature for 2 h. Second, the amino group of LF was esterified to the carboxyl group of HA using EDC as a coupling reagent. The resultant conjugates were then used as an emulsifier to prepare sesamol-loaded emulsions. We hypothesized that the covalent LF-HA conjugates would be better emulsifiers than either LF alone or simple LF-HA mixtures. Our main objective was therefore to better understand the impact of LF-HA conjugates on the formation and physicochemical stability of sesamol-loaded emulsions. The LF-HA conjugates synthesized in this study may have potential applications in the food and other industries as an effective biopolymer-based emulsifier.

## 2. Results and Discussion

### 2.1. SDS-PAGE Analysis

Gel electrophoresis was used to characterize the nature of the individual biopolymers and their conjugates. The SDS-PAGE patterns of LF, LF-HA mixtures, and LF-HA conjugates are shown in Figure 1. The band patterns for both LF-HA mixtures (lanes a and b) were similar to that of LF alone (lane c), indicating that protein cross-linking did not occur in these samples. On the other hand, the electrophoretic bands for the LF-HA conjugates (lanes d and e) migrated more slowly, which indicated that HA had covalently attached to the LF to form conjugates with higher molecular weights. Presumably, SDS disrupted the non-covalent bonds holding the LF and HA together in the electrostatic complexes, and so no higher molecular weight species were seen. On the other hand, SDS was unable to disrupt the covalent amide bonds holding the LF and HA molecules together. A similar phenomenon has been reported by He et al. [18], who showed that the formation of an amide bond between bovine serum albumin and chitosan could be achieved using EDC as a coupling reagent.

### 2.2. Structural Characterization of LF-HA Systems

#### 2.2.1. FTIR

The formation of LF-HA conjugates was further confirmed by FTIR analysis. Figure 2a shows FTIR spectra of lyophilized LF, LF-HA mixtures, and LF-HA conjugates. LF possessed characteristic bands at 3299 cm^−1^ (amide A band, representative of N–H stretching coupled with hydrogen bonding) and 1655 cm^−1^ (amide I, representive of C–O stretching/hydrogen bonding coupled with COO^−^). Compared to LF, the spectra of LF-HA conjugates showed a distinct difference at around 1669 cm^−1^, which was attributed to the formation of the ester bond or amide linkage [19]. Moreover, the FTIR spectra of the LF-HA conjugates showed a red shift at 3379 or 3410 cm^−1^, suggesting that the -OH or -NH_2_ group of LF may be involved in the reaction. The relatively large difference in the spectral change at 1669 cm^−1^(due to the C=O group) for the C(2:1) samples compared to the C(3:1) samples suggests that conjugate formation depended on the ratio of LF to HA used. This result was in agreement with previous findings suggesting that HA was grafted to the LF partially via the formation of a new amide bond [18,20].

#### 2.2.2. Fluorescence Spectra

Fluorescence spectroscopy measurements were used to provide information about changes in the local environment of the tryptophan residues in the protein [21]. As shown in Figure 2b, when excitated at 295 nm, LF exhibited a fluorescence emission maximum at 326 nm, and the fluorescence intensities of the LF-HA mixtures and conjugates were significantly lower than that of the pure LF, indicating that tryptophan may be involved in the binding between LF and HA [22].

The fluorescence intensity of the LF-HA C(3:1) was appreciably lower than that of the other samples and the absorption peak underwent a red shift (Figure 2b), which suggested that the extent of the covalent reaction depended on the ratio of reactants used. Moreover, our results indicated that the structure of the LF was changed after conjugation.

### 2.3. Proposed Formation Mechanism of LF-HA Conjugates

EDC is widely used as a carboxyl activating agent to promote the formation of amide bonds in proteins [18]. In our study, EDC was used to initiate the reactive carboxyl groups on HA, and NHS was used to increase the efficiency of the EDC coupling reaction. The formation of an amide bond between the carboxyl group of HA and a free amino group on LF was supported by the SDS-PAGE and FTIR measurements. A detailed mechanism for the chemical reaction between HA and LF through carbodiimide-mediated coupling is proposed in Figure 3. The carboxyl groups of hyaluronic acid were first activated with EDC and NHS in distilled water at pH = 4.5 for 2 h to produce semi-stable amine-reactive NHS-esters (products 1 and 2). The final reaction of hyaluronic acid with the LF amino group is very favorable and produces a stable amine linkage (final product 3 and intermediate 4). Finally, the unreacted molecules, intermediate 4, and NHS were removed by dialysis. Similar reactions using HA to form an amide bond in the presence of EDC and NHS have been reported previously [17].

### 2.4. Emulsion Particle Size and ζ-Potential Analysis

In this series of experiments, the impact of emulsifier type on the size and electrical characteristics of the particles in the sesamol emulsions was measured. As shown in Figure 4a, the mean particle diameter of the emulsions stabilized by the LF-HA mixtures and conjugates was always less than 550 nm, which is equal to or less than that of the LF-stabilized emulsions. This suggests that the LF-HA systems either promoted droplet disruption or inhibited droplet aggregation during homogenization. Previous studies have shown that emulsions containing smaller droplets are more stable to creaming and aggregation [23], and therefore the LF-HA systems should enhance emulsion stability. In addition, the polydispersity indices of all the LF-HA-emulsions were significantly smaller (*p* < 0.05) than those of the LF-emulsions, which suggested that a more uniform particle size distribution was achieved.

The electrical characteristics of the droplets in the emulsions impacted their stability and functional performance [24]. In general, the higher the absolute value of the ζ-potential, the greater the electrostatic repulsion, and the stronger the resistance to droplet aggregation. The ζ-potentials of the droplets in the LF, LF-HA M(2:1), and C(2:1) emulsions were negative, while those in the LF-HA M(3:1) and C(3:1) emulsions were close to zero (Figure 4b). The strong negative charge on the LF-coated droplets was surprising because this protein would be expected to be positively charged below its isoelectric point, which is around pH = 8. This may have been due to the adsorption of anionic impurities to the protein surfaces. We postulate that the droplets in the emulsions formulated from LF-HA M(2:1) and C(2:1) had a higher negative charge than those stabilized by LF alone due to binding of anionic HA molecules to positive patches on the surfaces of the LF molecules. We also propose that the ζ-potential was close to zero for the LF-HA M(3:1) and C(3:1) systems due to the fact that they had a higher LF content, which neutralized more of the negative charge on the HA molecules.

In summary, the polydispersity of the emulsion formulated from LF-HA C(2:1) was the lowest and the absolute value of the ζ-potential was the largest, which would be expected to lead to the best stability characteristics.

### 2.5. Changes in Physical State of Different Emulsions During Storage

It is well known that the composition and structure of the interfacial layer surrounding lipid droplets have a pronounced impact on their physicochemical stability [25]. The size and charge characteristics of the droplets in the different sesamol emulsions was therefore measured during storage (Figure 5). The size of the droplets in the emulsions stabilized by the conjugates scarcely changed during storage, but that of the other emulsions increased significantly, especially those formulated from the LF-HA mixtures (Figure 5a). The changes in the polydispersity index followed a similar trend to the particle size changes (Figure 5b). These results indicated that the emulsions stabilized by the LF-HA conjugates had better physical stability throughout storage. The fact that these emulsions initially had a ζ-potential close to zero suggests that the stability of the droplets was mainly a result of a strong steric repulsion between them. Presumably, the conjugates formed a thicker and stronger interfacial layer around the oil droplets. The poor stability of the emulsions formulated using the LF-HA mixtures may have been because of bridging flocculation, i.e., sharing of a single HA molecule between two or more droplets.

The trends in the ζ-potential values for the emulsions prepared using the different emulsifiers were fairly similar before and after storage, with the exception of the samples formulated with the LF-HA C(3:1). In this case, the droplets became much more negative after storage. The origin of this effect is currently unknown, but it may have been due to some leaking of excess LF molecules out of the interfacial region.

The visual appearance of different emulsions was recorded before and after storage at 60 °C for 10 days (Figure 5d). The photographs showed that the color and uniformity of the emulsions changed significantly after storage. All of the emulsions changed from white to brown, which is evidence of a chemical transformation of one of the components present. The emulsions prepared from the LF-HA mixtures were highly unstable to creaming and phase separation, while those prepared from the LF-HA conjugates exhibited excellent stability. The emulsions stabilized by LF alone also showed good stability. These results were consistent with the particle size results and suggested that conjugation of LF with HA can improve the storage stability of the emulsions.

### 2.6. Chemical Stability of Sesamol Emulsions

The effect of the different emulsifiers on the chemical stability of sesamol in the emulsions was studied in this series of experiments. Appreciable sesamol degradation was seen in all of the emulsions after one-week storage at 60 °C, and the extent of degradation was dependent on emulsifier type (Figure 6). The LF-HA conjugates were much more effective at inhibiting sesamol degradation than LF and LF-HA mixtures, suggesting that the conjugates had a protective role. A possible explanation for the ability of the conjugates to protect sesamol from chemical degradation was that they also protected the emulsions from physical breakdown. As a result, the sesamol remained inside the oil droplets, rather than being released into a separated oil phase. Further studies are needed to clarify the precise mechanism by which the LF-HA conjugates alter the degradation of sesamol in emulsions.

## 3. Materials and Methods

### 3.1. Materials

The bovine LF (purity > 98.92%) obtained from Westland Milk Products (Hokitika, New Zealand) was reported to contain 98.9 % protein, 0.6% moisture, and 0.5% ash. The sodium form of hyaluronic acid with a molecular weight of 9.8 kDa was purchased from FuRuiDa Pharmaceutical Company, Ltd. (Shandong, China). Sesamol, 1-(3-dimethylaminopropyl)-3-ethylcar-bodiimide hydrochloride (EDC), N-hydroxysuccinimide (NHS) and dialysis bags with a molecular mass cut-off at 12-14 kDa were purchased from the Sigma-Aldrich Chemical Company (St. Louis, MO, USA). Corn oil was purchased from Shandong Luhua Group Co. Ltd. (Shandong, China). All other chemicals and reagents were of analytical grade.

### 3.2. Preparation of LF-HA Mixtures and Conjugates

LF-HA mixtures and conjugates were prepared by physical mixing and physical mixing/EDC-cross linking, respectively. Briefly, HA, EDC, NHS, and LF solutions were prepared by dissolving the ingredients in distilled water (pH = 4.5) and stirring. The molar ratios of HA, EDC, NHS, and LF present in the final solutions were either 2:2:2:1 or 3:3:3:1, which was based on the results of a previous study [26]. These final solutions were prepared as follows, EDC and NHS were added to the HA solution, mixed well by magnetic stirring, and reacted at room temperature for 2 h. LF solution was then added drop wise to this mixture with vigorous stirring. The reaction mixture was allowed to stand at room temperature for 48 h, and the resulting system was then centrifuged at 8000 rpm and 4 °C in a high-speed centrifuge until it was no longer cloudy. The sample was then dialyzed using a dialysis bag to remove the intermediate and NHS. It was then freeze-dried using a lyophilizer (Alpha 1-2D Plus, Marin Christ, Germany) for 48 h as described previously [14]. The physical mixture was prepared in exactly the same manner at pH = 7.0, but without EDC and NHS. Finally, the prepared samples were stored at 4 °C. Conjugates of LF and HA at a molar ratio of 2:1 and 3:1 are referred to as LF-HA C(2:1) and C(3:1), respectively. Simple physical mixtures of LF and HA at a molar ratio of 2:1 and 3:1 are referred to as LF-HA M(2:1) and M(3:1), respectively.

### 3.3. SDS-PAGE

SDS-PAGE was performed using a discontinuous buffered system according to the method of Laemmli [27] to identify the molecular weight characteristics of the LF-HA conjugates. A vertical plate with a gel thickness of 1 mm, an operating voltage of 120 V, a 12.5% acrylamide separating gel, and a 4% stacking gel containing 0.1% SDS was used. The sample load per hole was 10 μL, and the marker sample load was 3.5 μL. After the end of electrophoresis, the gel strips were stained with Coomassie Brilliant Blue R-250 for 2 h, then decolorized with a solution composed of 10% methanol and 10% acetic acid. Gel images were scanned using a gel imaging system (Bio-Rad, Hercules, CA, USA).

### 3.4. Characterization of Particles LF-HA Systems

#### 3.4.1. FTIR

FTIR spectra were obtained using an infrared spectrometer (Vertex 70 FT-IR Spectrometer, Bruker, Germany) with 32 scans at a resolution of 4 cm^−1^ [28]. After the potassium bromide powder was ground, it was placed in an oven overnight, and the lyophilized sample was powdered and mixed with potassium bromide at a ratio of 1:100, and then ground into a uniform powder using an agate mortar. Measurements were performed in the mid-infrared region (4000–400 cm^−1^). Using potassium bromide as a blank, the spectra of each sample was acquired three times under the same conditions. The data were analyzed with an instrument software package (Omnicversion 6.1a, Thermo Nicolet Corp. Waltham, MA, USA).

#### 3.4.2. Fluorescence Spectrum Analysis

The fluorescent measurements were carried out using a fluorescence spectrometer (LS55, PerkinElmer, MA, USA) at room temperature. The concentrations of the samples used were 1 mg/mL. The scanning conditions were set to an excitation wavelength of 295 nm to selectively excite the tryptophan residues and an emission wavelength of 300~400 nm [29]. The excitation and emission slit widths were 5 nm and 3 nm, respectively, and distilled water containing no sample was used as a blank control.

### 3.5. Preparation of Sesamol Emulsions

Oil-in-water emulsions were prepared according to a method described previously [30]. An oil phase was prepared by dissolving sesamol (2 wt%) in corn oil heated to 100 °C for 30 s. Aqueous phases were prepared by dissolving LF, LF-HA mixtures, or LF-HA conjugates (0.5 wt%) in distilled water and stirring for 12 h. The oil phase (5 wt%) and aqueous phase (95 wt%) were then used to make sesamol emulsions. Under magnetic stirring, the oil phase was slowly added to the aqueous phase, and sheared at 10,000 r/min for 6 min using a high-shear mixer (Ultra Turrax T25, IKA, Staufen, Germany) to form a coarse emulsion. Then, the coarse emulsions were further homogenized at 50 MPa through a high-pressure homogenizer (ATS Engineer Inc., Shanghai, China) for three cycles. The emulsions prepare were then quickly transferred to glass bottles, sealed, and stored at 4 °C until analysis.

### 3.6. Measurements of Droplet Size and ζ-potential of Sesamol Emulsions

Dynamic light scattering was used to measure the mean particle diameter and particle size distribution of the sesamol-loaded emulsions (Zetasizer Nano-ZS90, Malvern Instruments, Worcestershire, UK) at a fixed detector angle of 90°. The sample was diluted 500-fold with distilled water before the measurement to avoid multiple scattering effects. Results are expressed as the mean particle diameter (z-average) and polydispersity index (PdI). Each sample was measured in triplicate.

Electrophoresis was used to measure the electrical characteristics of the particles in the diluted sesamol emulsions (Nano-ZS90, Malvern Instruments, Worcestershire, UK) using the Smoluchowski equation to convert the measured electrophoretic mobilities into ζ-potential values.

### 3.7. Storage Stability of Sesamol-Loaded Emulsions

Aliquots of the sesamol-loaded emulsions (5 mL) were transferred into test tubes, and then their physical and chemical stability were determined periodically during storage. The physical stability of the emulsions was determined by placing them in sealed bottles stored at 60 °C, and then measuring changes in their mean particle diameter, polydispersity index, ζ-potential, and visual appearance throughout storage. The chemical degradation of the sesamol in the emulsions was determined by measuring changes in the sesamol concentration in the emulsions using a method described previously [31].

Initially, 1 mL samples were first extracted by adding 2 mL ethanol and 3 mL of n-hexane. The extraction was repeated three times and the total sesamol content was determined in the combined *n*-hexane. After appropriate dilutions with *n*-hexane, sesamol was determined colorimetrically at 298 nm [32] on a Shimadzu UV 1240 spectrophotometer (UV-1240 Shimadzu, Tokyo, Japan). The concentration of sesamol was obtained by referring to a standard curve of sesamol prepared under the same conditions. The sesamol content in the emulsions during storage was normalized to the initial amount.

### 3.8. Statistical Analysis

All experiments were conducted in triplicate and data were subjected to analysis of variance (ANOVA) using the SPSS 12.0 package (SPSS Inc., Chicago, IL, USA). Significant differences of means were determined by the Duncan’s multiple range test at the significance level of 5%.

## 4. Conclusions

LF-HA complexes were successfully prepared using a combination of physical mixing and covalent cross-linking. Gel electrophoresis measurements showed that the molecular weight increased after conjugation, while spectroscopic analysis showed that conjugation changed the structure of the protein. The LF-HA conjugates were able to effectively improve the physical stability of the emulsions, as well as to inhibit the chemical degradation of sesamol during storage. This study suggests that LF-HA conjugates may be useful emulsifiers to improve both the physical and chemical stability of emulsions. However, further research needs to be performed to clarify the stabilization mechanism of the LF-HA conjugates.

## Figures and Tables

**Figure 1 molecules-23-03291-f001:**
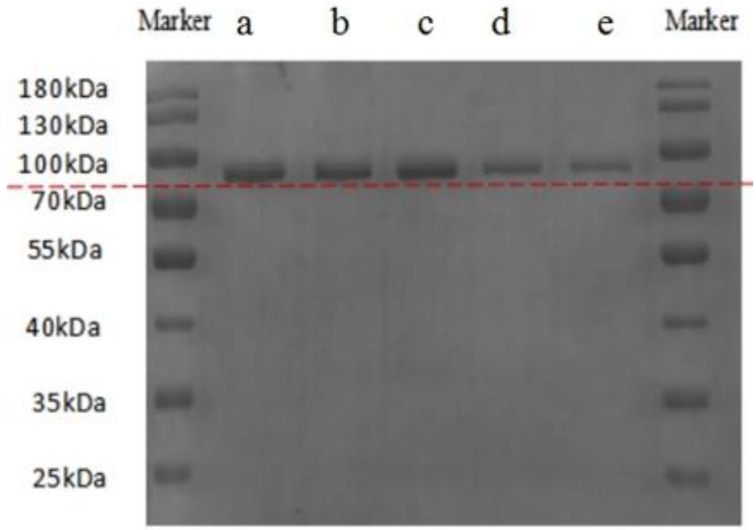
SDS-PAGE of LF and LF-HA complexes, a: LF-HA M(2:1); b: LF-HA M(3:1); c: LF; d: LF-HA C(2:1); e LF-HA C(3:1).

**Figure 2 molecules-23-03291-f002:**
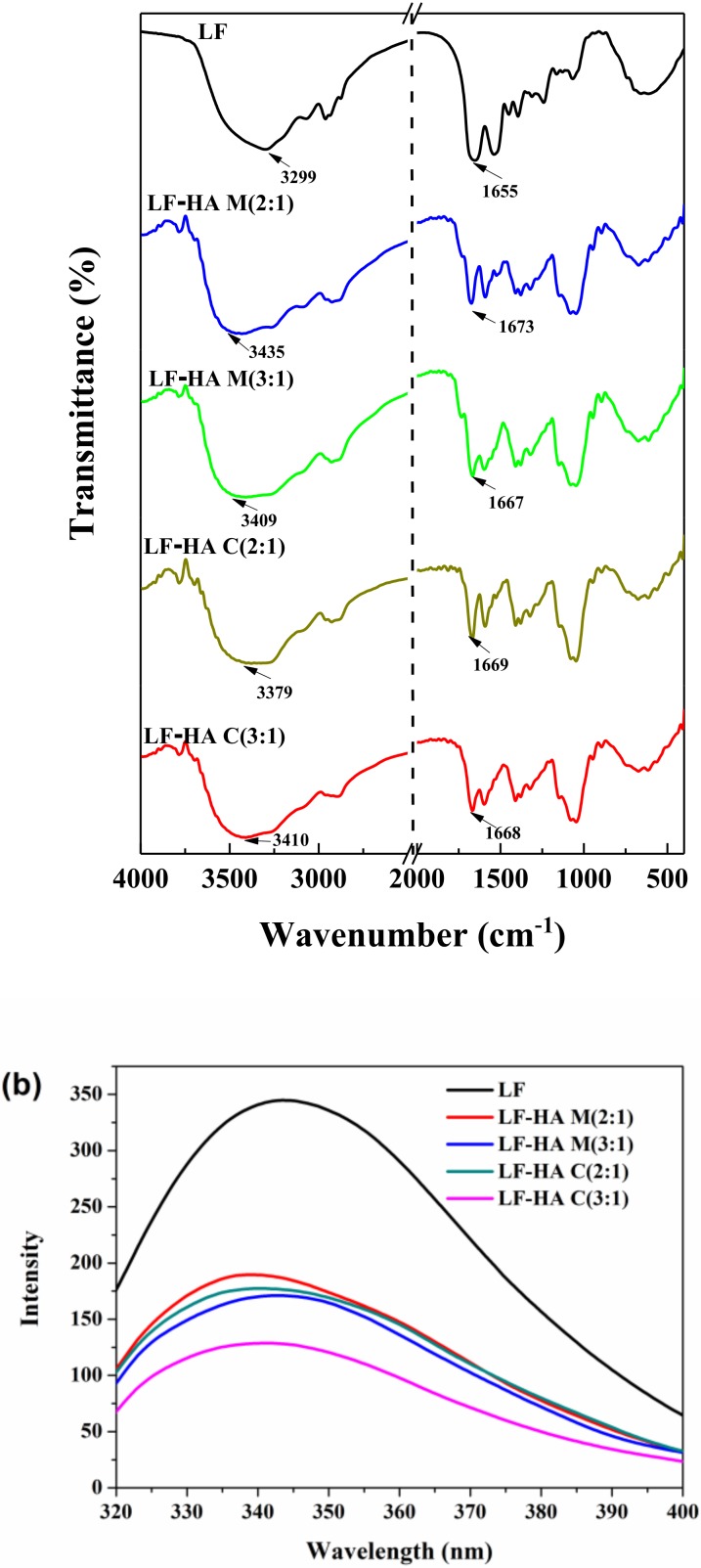
Fourier transform infrared spectra (**a**) and fluorescence spectra (**b**) of LF and LF-HA complexes.

**Figure 3 molecules-23-03291-f003:**
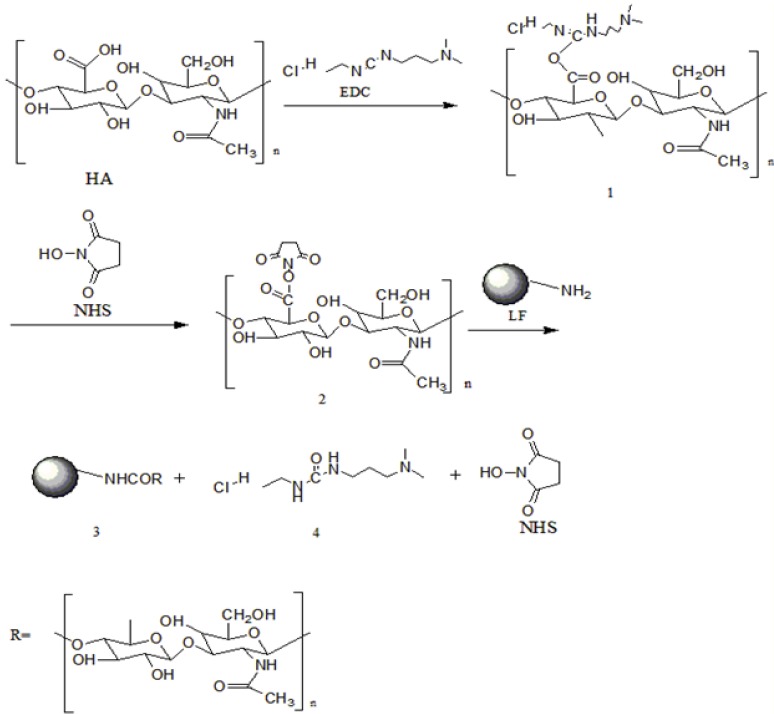
Reaction pathway of HA conjugated with LF via conjugating agents.

**Figure 4 molecules-23-03291-f004:**
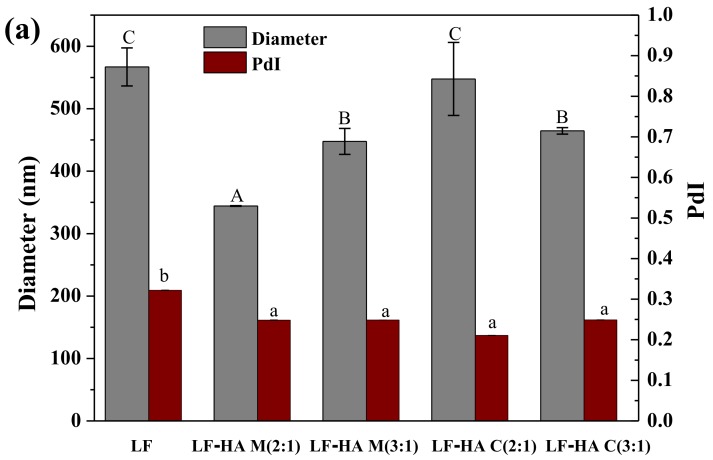
Particle size, PdI (**a**) and zeta potential (**b**) of different sesamol emulsions stabilized by LF and LF-HA complexes. Bars with different letters are significantly different from each other (*p* < 0.05).

**Figure 5 molecules-23-03291-f005:**
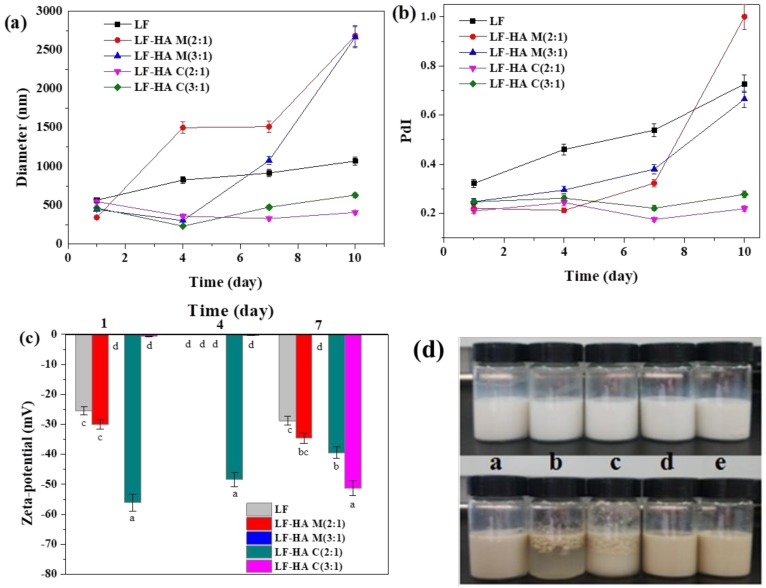
(**a**) Changes in particle size, (**b**) PdI, (**c**) zeta-potential and (**d**) morphology of different sesamol emulsions stabilized by LF and LF-HA complexes during storage at 60 °C. The letters in (**d**) are emulsions stabilized by, a: LF; b: LF-HA M(2:1); c: LF-HA M(3:1); d: LF-HA C(2:1); e: LF-HA C(3:1). Bars with different letters are significantly different from each other (*p* < 0.05).

**Figure 6 molecules-23-03291-f006:**
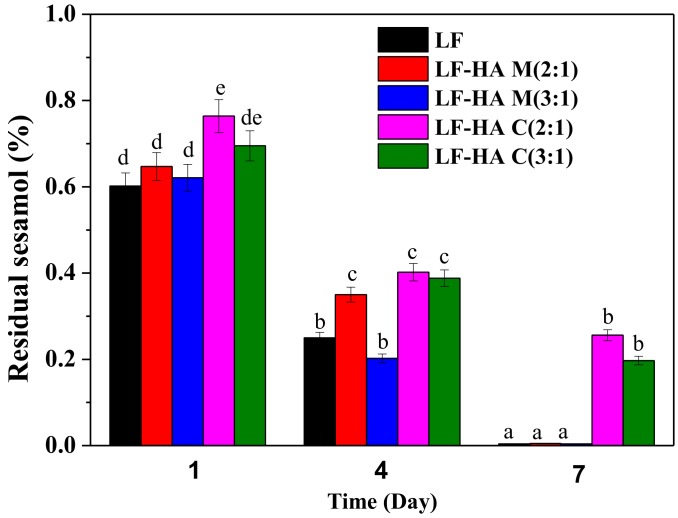
Changes in sesamol content in different emulsions stabilized by LF and LF-HA complexes during storage at 60 °C. Bars with different letters are significantly different from each other (*p* < 0.05).

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
