# Peer review of "Formation and Characterization of Lactoferrin-Hyaluronic Acid Conjugates and Their Effects on the Storage Stability of Sesamol Emulsions"

_molecules, 2018, doi:10.3390/molecules23123291_

Round 1

Reviewer 1 Report

The work is very well designed and the methods are adequate employed.

I strongly encourage the authors to check the entire manuscript for typo and grammar errors. A native english speaker could help in that task.

1. Some more details about the emulsification process is necessary to better understand the mechanism of stabilization.

2. The authors determine the physical stability of the emulsions, but should be interesting to present the droplet size distribution of emulsions.

3. The mechanism of physical stabilization of emulsion should be explained. Could be a Pickering emulsion?

4. The authors should mention how this study can contribute to future studies that can, eventually, lead to a commercial formulation.

Author Response

Reviewer #1: The work is very well designed and the methods are adequate employed.

I strongly encourage the authors to check the entire manuscript for typo and grammar errors. A native english speaker could help in that task.

Response: Thank you for your nice comments. We have improved the English of our manuscript by a colleague who is well-versed in English, and we revised the manuscript in detail and marked in red in the revised manuscript.

Specific comments:

1. Some more details about the emulsification process is necessary to better understand the mechanism of stabilization.

Response: Coarse oil-in-water emulsions containing 5% (w/w) corn oil phase and 95% (w/w) aqueous phase were formed using a high-shear mixer (Ultra Turrax T25, IKA, Germany) operated for 6 min at 10,000 rpm. These emulsions were then passed through a high-pressure homogenizer (ATS Engineer Inc., Shanghai, China) for three cycles to form the final emulsions.

2. The authors determine the physical stability of the emulsions, but should be interesting to present the droplet size distribution of emulsions.

Response: Thank you for your good suggestions. To reflect the particle size distribution, we determined the PdI values of all samples, which stands for the width of particle size distribution. The PdI values of initial emulsions stabilized by LF-HA complexes were found to be in the range of 0.2~0.25, indicating narrow particle size distribution. The changes in PdI of different sesamol emulsions during storage were discussed in the revised manuscript. 

3. The mechanism of physical stabilization of emulsion should be explained. Could be a Pickering emulsion?

Response: Pickering emulsions were stabilized by colloidal solid particles. Based on our results, the emulsions in this study were stabilized by liquid emulsifier of LF-HA complexes and therefore they were not pickering emulsions. LF-HA conjugates performed better stability than LF-HA mixture and single LF. The main reason is that bridging flocculation occured in the emulsions formulated using the LF-HA mixtures, i.e., sharing of a single HA molecule between two or more droplets, but for LF-HA conjugates, the increased steric hindrance and electrostatic repulsion by HA prevented bridge flocculation and therefore contributed the better stability of the droplets.

4. The authors should mention how this study can contribute to future studies that can, eventually, lead to a commercial formulation.

Response: Thank you for your nice comments. In this study we primarily explored the stability of the three different emulsions, and found LF-HA conjugate was spurious than LF-HA mixture, but further research needs to be performed to clarify the stabilization mechanism and toxicological evaluation should be carried out to guarantee the safety the formulation.

Reviewer 2 Report

In this manuscript, Liu and coworkers reported a new formulation for the encapsulation and storage of sesamol. The article is highly interdisciplinary in nature covering aspects of polymer synthesis, nanoparticle fabrication, as well as physiochemical characterization. The paper represents a useful contribution in the field of food science. This manuscript would be a more comprehensive work and suitable for publication if some revisions could be made.

Specific comments:

1. Gel electrophoresis, IR and fluorescence spectra results were shown to validate the LF-HA conjugation. They also proposed the reaction pathway of LF-HA conjugation in Figure 3. The authors should provide the NMR-spectrum of each compound in Figure 3 to validate the conjugation of NHS ester or even LF on carboxyl group (C-5). NMR is a more direct evidence of conjugation.

2. The authors claimed that the optimum emulsion stability was at a LF-to-HA molar ratio of 2:1. However, they only showed two formulations with ratio of 2:1 and 3:1 based on a previously study which showed the conjugation of chitosan and gallic acid. The authors should screen more formulations or provide more evidence to validate why 2:1 is the best formulation for current LF-HA system.

3. Figure 4 showed that the nanodroplets lost 40% of sesamol after 1 day. Can these authors further explain the observed burst release? It results from free sesamol or just the gradient-driven diffusion.

4. The authors should also add the TEM images of the obtained nanodroplets.

Author Response

Reviewer #2: Comments and Suggestions for Authors

In this manuscript, Liu and coworkers reported a new formulation for the encapsulation and storage of sesamol. The article is highly interdisciplinary in nature covering aspects of polymer synthesis, nanoparticle fabrication, as well as physiochemical characterization. The paper represents a useful contribution in the field of food science. This manuscript would be a more comprehensive work and suitable for publication if some revisions could be made.

Specific comments:

1. Gel electrophoresis, IR and fluorescence spectra results were shown to validate the LF-HA conjugation. They also proposed the reaction pathway of LF-HA conjugation in Figure 3. The authors should provide the NMR-spectrum of each compound in Figure 3 to validate the conjugation of NHS ester or even LF on carboxyl group (C-5). NMR is a more direct evidence of conjugation.

Response: Thank you for your nice comments. Your advice was meaningful and constructive, and we have supplemented the results of the NMR as bellow.

Fig. 2 1H NMR spectra of LF and LF-HA complexes in D2O.

As shown in the 1H NMR data (Fig. 2), the solvent peak of D2O at 4.65 ppm was used as the reference. The NMR spectrum of lactoferrin did not showed characteristic peaks in D2O above 2.5 ppm, while the NMR spectrum of LF-HA C(2:1) showed a new characteristic peak at 3.0 ppm, which was not found in the spectra of both HA and LF-HA M (2:1), confirming the LF-HA conjugates were prepared by the amide linkages between amino groups of the LF and carboxylic groups on HA.

2. The authors claimed that the optimum emulsion stability was at a LF-to-HA molar ratio of 2:1. However, they only showed two formulations with ratio of 2:1 and 3:1 based on a previously study which showed the conjugation of chitosan and gallic acid. The authors should screen more formulations or provide more evidence to validate why 2:1 is the best formulation for current LF-HA system.

Response: Thank you for your good comments. Based on a previous study, conjugates were prepared by varying the ratio of amino group to carboxyl group (Pasanphan, & Chirachanchai, 2008). In addition, a review by Lee, et al. (2017) found that increasing the amino : carbonyl ratio from 1:1 to 1:2 or higher resulted in a increase in the degree of glycation. This can be explained by the extra carbonyl groups that are available for conjugation with free amino acid. Furthermore, the large molecular size of the polysaccharide enables the formation of thick adsorption layer surrounding emulsion droplet, providing stearic hindrance effect that prevents droplet from aggregation and flocculation.

References:

Lee, Y. Y., Tang, T. K., Phuah, E. T., Alitheen, N. B. M., Tan, C. P., & Lai, O. M. (2017). New functionalities of Maillard reaction products as emulsifiers and encapsulating agents, and the processing parameters: a brief review. Journal of the Science of Food and Agriculture, 97(5), 1379-1385.

Pasanphan, W., & Chirachanchai, S. (2008). Conjugation of gallic acid onto chitosan: An approach for green and water-based antioxidant. Carbohydrate Polymers, 72(1), 169-177.

3. Figure 4 showed that the nanodroplets lost 40% of sesamol after 1 day. Can these authors further explain the observed burst release? It results from free sesamol or just the gradient-driven diffusion.

Response: Sesamol has been known as a lipid antioxidant since 1950s, and as an antioxidants, sesamol was not stable and can be oxidative-decomposed. During the preparation of the emulsion, there is a certain loss of sesamol due to the action of heating and oxygen. LF-HA conjugates exhibited best protection to avoid sesamol degradation.

4. The authors should also add the TEM images of the obtained nanodroplets.

Response: As you suggested, the results of the TEM images of the droplets have been added and presented as follows:

Fig. 3 Transmission electron micrograph of different sesamol emulsions stabilized by LF and LF-HA complexes.

TEM has been scarcely employed to examine the microstructure of emulsions because the sample preparation process will destroy the structures of the emulsion. As shown in figure 3, TEM images of LF and LF-HA mixture stabilized-emulsion were similar, but the conjugates-stabilized enulsions exhibited diffuse droplet structure due to thicker interfacial layer and the crosslinked emulsifier. The TEM images further confirm the difference between different emulsifiers.

Thank you again for your good comments. We hope the revised manuscript will be suitable for publication in Molecules.

Yours sincerely,

Fuguo Liu

Beijing advanced innovation center for food nutrition and human health, Beijing Technology and Business University, Beijing, People's Republic of China

Round 2

Reviewer 2 Report

In the revised manuscript, the authors successfully addressed most of my concerns.

Appropriate references were added to support the molar ratio used in current study. NMR spectra were included to validate the conjugation of LF-HA.

The revised manuscript is much clear and leads to more solid science. I believe current manuscript is ready for publication.